# Does Entrepreneurial Financial Support Guarantee New Ventures' Performance via Competitive Advantage and Innovation? Empirical Answers from Ho Chi Minh City Region, Vietnam

Quoc Hoang Thai and Khuong Ngoc Mai *

School of Business, International University, Vietnam National University Ho Chi Minh City (VNU-HCM), Linh Trung Ward, Thu Duc District, Ho Chi Minh City 700000, Vietnam; thai.quoc2611@gmail.com
* Correspondence: mnkhuong@hcmiu.edu.vn

**Abstract:** This research applied the resource-based view (RBV) theory to examine the effects of entrepreneurial financial support on new ventures' performance via competitive advantage and innovation. A questionnaire survey was performed to collect data from 315 entrepreneurs of new ventures in the Ho Chi Minh City region. A quantitative method was applied, and partial least squares structural equation modeling (PLS-SEM) was utilized to confirm the significant relationships among variables of the research model. The findings demonstrated that all financial and operational performances of new venture constructs in this research model were completely supported. Furthermore, entrepreneurial financial support had the strongest direct and indirect effects on firm innovation and competitive advantage, while firm innovation and competitive advantage fully mediated relationships between entrepreneurial financial support and firms' financial and operational performance. Hence, this research solved extant debates in the literature, concurrently enhancing the RBV theory in the entrepreneurship context. In the post-COVID-19 pandemic era, it offers novel insight for governors and other stakeholders to build an efficient financial support system, while providing entrepreneurs with strategies to leverage said system to develop innovation for achieving better competitive advantage, leading to higher firm performance.

**Keywords:** entrepreneurial financial support; firm performance; firm financial performance; firm operational performance; firm innovation; firm competitive advantage; new ventures; entrepreneurship; COVID-19 pandemic

## 1. Introduction

Entrepreneurship in Vietnam has undergone strong development among the government, organizations, and individuals because it is positively linked with economic expansion and national evolution. The number of new registered enterprises in 2022 was 62,961 organizations, reaching the highest rate in history, with a registered capital equal to VND 761,035 billion [1]. Ho Chi Minh City (HCMC) is a nascent and energetic territory that draws numerous distinct forms of enterprise to operate [2]. It was the economic hub of Vietnam, contributing the highest percentage of GDP (15.5 percent), in 2022 [3], contributing significantly to entrepreneurship and the economy of Vietnam [4]. HCMC has enforced the project "Assistance policies on creative and innovative entrepreneurial ecosystem in HCMC period 2021–2025" to develop an innovative entrepreneurial ecosystem for promoting new ventures [5]. However, starting in 2020, fluctuations emerged within the community and the economic environment because of the COVID-19 pandemic [6]. COVID-19 generated an uncertain and chaotic business climate and adversely influenced the long-term sustainability of new ventures, intimidating the sustainable business networks and economies of various regions [7]. The availability of entrepreneurial financial support (FIN) like credit,

bank loans, and due payments was progressively restricted because of the COVID-19 pandemic crisis [8], leading to adverse impacts on new ventures [9]. Overall, 90 percent of firms were critically damaged, with various firms restricting and pausing operations or even declaring bankruptcy [10]. It was claimed that the number of firms ceasing operations was large, exceeding 100,000 firms, whereas micro- and small-sized enterprises (MSMEs) and new ventures were the most damaged by the COVID-19 pandemic. In addition to that, the percentage of successful startups was 5 percent, while startups that experienced losses accounted for 37 percent [11]. These pessimistic consequences can be traced back to the negative circumstance of external resources, FIN [6,9,12–14], and internal resources, INO [15–17]. To overcome these issues, enterprises consider the competitive advantages of sustainability in business and apply innovation that is the core of sustainable value creation of new ventures [7,18–22]. In 2018, Vietnam was ranked 39th out of 54 in terms of its entrepreneurial ecosystem [23]. In addition, developing ventures faced many challenges in accessing finance and capital, which are limited in Vietnam [24], leading to limited admission to angel investors because of the presence of separation between the ancient economy and the modern one [25]. In modern businesses, sustainability is the most important element for businesses to enhance firm operational performance (BOP); however, the ancient economy focused less on innovation and sustainability in businesses with competitive advantage [26] because of large financial investments. For business competitive advantage and innovation [15] in enterprises, BOP has to wake up to this reality [27]. In addition to that, investment capital for entrepreneurship in Vietnam was still limited compared with other nations, which did not adapt to the demands of new ventures in Vietnam [28]. Regarding INO, the innovation index of entrepreneurial activities in Vietnam only reached 13.9 percent, ranking 48th among 54 nations [23]. The World Bank [29] demonstrated that the competencies of national and organizational science, technology, and innovation were still weak, nascent, and fragmented. Overall, 97 percent of Vietnamese enterprises have not innovated effectively to pursue worldwide technology trends; thus, INO will become a challenge for new ventures wishing to enhance their organizational sustainability [30]. Moreover, there is still a lack of favorable policies for creativity and innovation in Vietnam encouraging firms and private investors to engage in research, creative and innovative activities, and applications of new science and technology, meaning that new Vietnamese ventures rarely adapt, utilize, and improve new technology and innovation [31]. Thus, the competitiveness of Vietnam is still restricted and low due to the impact of both external (business climate, financial resources, etc.) and internal factors (management competencies, financial competencies, firm innovation, etc.) [32]. New ventures in Vietnam face several barriers and challenges including restricted venture capital, limited access to finance, and outdated equipment and technology and, thus, have less competitive advantages compared to other regions in Asia [33], resulting in deficient firm performance [34] while facing the COVID-19 pandemic compared with regional and global productivity standards [35]. To solve these grand challenges, new ventures in Vietnam have prioritized innovation [15] and built sustainable competitive advantage benefiting communities, governments, and the natural environment by developing strategies and business models that are agile and asset-focused. By exploiting eco-friendly products and services using renewable energy, locally sourced, or alternative materials [18,26,36,37], new ventures in Vietnam drive this transformation as a sustainable source to succeed over the long run, which can adjust or respond to their BOP [27] in response to increasing future costs that can influence their firm financial performance (FiPer) [18]. Therefore, exploring the antecedents of firm performance of new ventures in the HCMC region, Vietnam, during the COVID-19 pandemic and post-COVID-19 pandemic phases is a crucial issue that is investigated in this research because of their effects on the long-term survival and sustainability of new ventures, national economic growth, and sustainable nationwide progress [38]. Due to the above issues embedded in Vietnamese entrepreneurship, this research concentrates on the FIN and firm innovation (INO) as external and internal mechanisms that help new ventures in creating sustainable firm competitive advantage (FCA), turning into improved firm performance, to deal with

COVID-19 pandemic effects. In addition to that, it is urgent and necessary to conduct this research in order to propose valuable strategies to improve these factors that determine the long-term survival of new sustainable ventures because they help new ventures to enhance their firm performance and develop the national economy of Vietnam. Because sustainable development includes the three primary factors economic development, environmental protection, and social well-being elements [39]—which create opportunities for competitive advantage, improved performance of new ventures, innovation, and national economic growth—it transforms into sustainable entrepreneurship and national sustainable economic recovery by inciting a revolutionary change in the way we approach the essential facets of sustainable development.

According to the resource-based view (RBV) theory, the sustainable FCA and firm performance of an organization are dependent on the organizational possession of both external and internal resources and capabilities that are valuable, rare, inimitable, and non-substitutable [40,41]. Previous studies on FIN focused on external resources that significantly and positively contribute to FiPer and BOP across countries [6,9,12–14]. Due to the COVID-19 pandemic, the availability and adequacy of FIN are essential external resources for new ventures in achieving a better competitive position in the market and enhance firm performance [6]. Specifically, Jayeola et al. [9] demonstrated that the external resources obtained through FIN, which is acknowledged as an external mechanism by the RBV theory, facilitated more rapid economic growth for enterprises in gaining a stronger FCA and consequently enhanced new ventures' firm performance during the COVID-19 pandemic. FIN, through COVID-19 financial assistance, positively affects sustainable FiPer and the economic survival/recovery of MSMEs [14]. Because of issues related to the COVID-19 pandemic, enterprises interacted with their environment through online channels to access financial resources and become proactive in their operations to increase their BOP [13]. Following the ease of COVID-19 infection, FIN also stimulated the labor productivity and organizational productivity of new ventures in the early stage of economic recovery [12]. In addition to that, by using FIN, new ventures can develop their INO through enhancing investment in products and processes and mobilizing finance to facilitate firms' patenting activity [42–44]. Prior investigations also identified INO as another crucial factor, acknowledged as an internal mechanism by the RBV theory, that positively influenced new ventures' firm performance [15–17]. INO is the trademark of entrepreneurs and their new ventures, especially when they act toward uncertain circumstances such as the COVID-19 pandemic. Therefore, they utilize an innovative perspective to produce essential products and technological skills to improve their firm performance [6]. Hence, during the COVID-19 pandemic, the development of INO helped new ventures to generate innovative procedures, which offered novel products and services and promoted their growth in the market, thus positively facilitating both FiPer and BOP [45,46]. Furthermore, previous studies also showed that new ventures develop their FCA by utilizing the aforementioned external and internal mechanisms because they can obtain valuable and necessary resources to minimize costs and can produce unique products and services to achieve a competitive advantage to expand their market [47–49]. FCA is also another essential antecedent of new ventures' firm performance [18–22]. Because of the outburst of the COVID-19 pandemic, superlative challenges for new ventures emerged and, thus, the development of FCA was considered a useful strategy for new ventures' survival during the COVID-19 pandemic by improving their firm performance [50]. Since the COVID-19 pandemic disproportionately affected small- and medium-sized businesses (SMEs), the possession of FCA enhanced new ventures' firm performance by creating exceptional values for customers, improving their BOP [51]. In addition, by having higher levels of FCA, firms could exploit business opportunities and neutralize competitor threats, resulting in a higher firm performance during the COVID-19 pandemic [49,52].

Despite the fact that there are several studies in the literature that have researched the impacts of FIN on the firm performance of new ventures, there are still research gaps in the theoretical context due to the following reasons: Firstly, despite various investigations

confirming the positive effects of FIN on new ventures' firm performance [6,13,14], the influences of FIN on new ventures' firm performance are mixed because extant studies have also concluded that FIN does not unveil a direct statistically significant impact [9,53] or even has a negative impact [54,55] on the firm performance of new ventures, leading to debates in the literature. Secondly, the association between INO and firm performance has also generated similar debates by demonstrating mixed results [56]. Li and Atuahene-Gima [57] conducted a review on the link between INO and firm performance and deemed the evidence mixed and contrary, causing conflicts in the literature. Thirdly, there are two isolated streams of research in the literature; the first stream analyzes the effects of FIN on FCA and firm performance [13,14,47,58], while the second stream separately investigates the impacts of INO on FCA and firm performance [15,45,46,48,49]. Therefore, there is a lack of research that integrates the two streams due to the impacts of both internal and external mechanisms on FCA, which stimulates new ventures' firm performance [9]. Finally, recent systematic literature reviews on entrepreneurial ecosystems [59,60], encompassing FIN as a crucial domain, have been conducted to synthesize evolutionary trends and important content addressed in this research area, proposing current research gaps that should be investigated in further studies. Because the various studies that examined this phenomenon utilized a descriptive approach [15,60] for which findings were hard to generalize, it is necessary to conduct an empirical study (by creating and validating new appropriate measurements of FIN) and, thus, investigate the causal relationships of FIN in order to broaden the knowledge in the field of sustainable entrepreneurship.

To bridge these research gaps, the need for a study that integrates and investigates the two aforementioned research streams to fully improve the RBV theory emerged as an urgent concern in the entrepreneurship literature [9]. This research aims to analyze the influences of FIN on new ventures' firm performance through the mediating roles of FCA and INO, contributing significantly to the literature due to the following rationales: Firstly, this research utilizes the RBV theory to understand the roles of FIN and INO in achieving sustainable FCA and superior firm performance of new ventures, resolving extant debates related to FIN–firm performance links [9,53–55] and INO–firm performance associations [56,57]. Secondly, this research consults the work of Jayeola et al. [9] to propose that FIN captured by new ventures is utilized to develop internal competing resources and that capabilities—INO—could first stimulate new ventures in obtaining their FCA before improving firm performance in terms of FiPer and BOP. Thus, a new dual sequential mediation of INO and FCA is proposed to offer more comprehensive insight into the relationships between FIN and new ventures' firm performance. It also emphasizes a realistic and complicated mechanism underlying these relationships, in opposition to an uncomplicated mediation of sustainable FCA analyzed in extant studies to fully understand the RBV theory. Hence, this research builds upon extant studies by investigating a comprehensive picture of mediating roles of FCA and INO in the relationships between FIN and new ventures' firm performance. This issue is an ongoing debate and has not been fully examined in the literature, which points to our research's novelty and differentiation. Furthermore, by utilizing the most common measurement scales demonstrated in extant systematic literature reviews on entrepreneurial ecosystems, this research fulfils the demands of these systematic literature reviews [59,60] because it generates and validates a comprehensive measurement of FIN, identifying extensive external financial sources so that new ventures can conduct quantitative research and enhance causal relationships of FIN, thus strengthening the literature on sustainable entrepreneurship. This research aims to solve the following research questions:

RQ1. To what extent does entrepreneurial financial support directly and indirectly affect new ventures' firm performance, competitive advantage, and innovation?

RQ2. Do firm innovation and competitive advantage mediate the relationships between entrepreneurial financial support and new ventures' firm performance?

To answer the above research questions, we performed and analyzed a questionnaire survey of 315 entrepreneurs of new ventures operating in the HCMC region to investigate

the effects of FIN on new ventures' FiPer and BOP via the mediating roles of FCA and INO. Specifically, we focused on FIN as an external mechanism that is utilized by entrepreneurs and their new ventures to develop their firm innovation (INO) as an internal mechanism that helps new ventures to create a sustainable firm competitive advantage (FCA), resulting in improved firm performance to deal with the lasting effects of the COVID-19 pandemic. The findings demonstrated that all FiPer and BOP of new venture constructs in this research model were completely supported. Furthermore, FIN had the strongest direct and indirect effects on INO and FCA, while INO and FCA fully mediated relationships between FIN and new ventures' FiPer and BOP. The next section introduces the theoretical background and hypotheses' development. The subsequent sections consist of methodology, data analysis and results, discussion, and conclusions.

## 2. Theoretical Background and Hypotheses' Development

### 2.1. Resource-Based View Theory

RBV theory was developed as a crucial theoretical viewpoint applied to demonstrate tenaciousness in differences in performance across firms [61]. According to the RBV theory, the sustainable FCA and outcomes of an enterprise are the result of the possession of unique organizational resources that are valuable, rare, inimitable, and non-substitutable [40,41,62]. By adopting the RBV theory, this research analyzes FIN as an external mechanism and INO as an internal mechanism and their capacity to stimulate new ventures in obtaining a sustainable FCA and enhanced firm performance. Regarding the external mechanism, the RBV theory reveals that enterprises can strengthen their resources through obtaining additional resources from an external system like FIN systems [63]. The RBV theory argues that externally available resources can affect new ventures' firm performance [64,65]. Hence, superior firm performance is a result of external resources, which is demonstrated as FIN [66,67]. Regarding the internal mechanism, capacity explains the formation and rearrangement of resources to foster productivity and accomplish strategic objectives [68]. Hence, Camisón and Villar-López [69] claimed that the establishment of INO in an enterprise can be characterized as an actual source of FCA [70], which causes an improvement in firm performance [71] because it generates new products or services, technologies, organizational design, and management methods. The RBV theory was also utilized in the work of Prange and Pinho [72] to view INO as internal resources that express the consistent arrangement of organizational resources and capabilities to promote novel productions and markets [73], resulting in enterprises' heterogeneity and exceptional performance [69,74].

### 2.2. Firm Performance

Firm performance refers to a firm's ultimate result, comprising managerial effectiveness and elements beyond organizational management [75]. Furthermore, it is the overall level of satisfaction among all participants when they collect and transform inputs into outputs in an effective method [76]. It estimates what stimulates a firm to generate intended outcomes in terms of both means and ends [77]. Considering existing research on firm performance, various measurements have been applied. However, this research followed the framework of Venkatraman and Ramanujam [78] to clarify firm performance as a broad concept encompassing two smaller domains including FiPer and BOP. FiPer relates to the method that illustrates the overall performance of a firm, which is measured as the profitability (indicated by ratios such as return on assets, return on sale, and return on equity) [18,78]. BOP demonstrates long-term objectives and development capabilities, combining measures like new product establishment, product quality, manufacturing value-added, productivity, growth, satisfaction of stakeholders, and efficiency [27,78].

### 2.3. Entrepreneurial Financial Support

FIN focuses principally on the capacity to obtain financial resources provided to entrepreneurs [79]. It identifies those financial organizations that handle tasks of entrepreneurs' funding, encompassing "micro-loan, angel investors, zero-stage venture cap-

ital, venture capital funding, private equity, public capital markets, and debt" [80]. FIN is also regarded as sources of funding and finance including "friends and family, angel investors, private equity, venture capital, and access to debt" [81,82]. In addition, FIN refers to all components relevant to any forms of financing, encompassing "public subsidies or helps, informal investment, banks, credit, microcredits, venture capital, and others" [83]. This research uses the RBV theory to analyze FIN as the availability of external financial resources for new ventures [84], including local investors, local communities, friends and family, banks, bankers, funding programs, and government subsidies [79,82,83].

### 2.4. Firm Innovation

INO is a novel concept, technique, or mechanism that can be obtained through offering a novel production, organizational design, management process, or modification in organizational culture [85]. INO is acknowledged as a combination of managerial activities, focusing on the renewal and development of organizational designs, operations, and methods with the aim to enhance organizational targets [86]. INO also illustrates the competencies to create and utilize novel concepts or attitudes. It is crucial for strengthening organizational outputs, resulting in high performance [87]. INO refers to a crucial approach through which organizations can aim to adapt novel equipment, methods, and administrative procedures associated with other innovative activities, enabling organizations to create an essential contribution to innovation procedures [88]. This research utilizes the RBV theory to clarify INO as the internal operation of generating a novel product or process that encompasses creations and activities necessary to transform an idea or concept into a final pattern, improving the firm performance of an organization [15].

### 2.5. Firm Competitive Advantage

FCA is obtained in the circumstance that an organization's strategy generates values that competitors cannot imitate or exploit. Organizations have to develop differentiation strategies to acquire FCA by having exclusive resources and abilities to focus on four major facets including value, rareness, imitability, and organization [40]. Peteraf and Barney [62] developed the following explicit definition of FCA: a firm possesses an FCA in the case that it has abilities to generate more economic value, which is the distinction between consumers' perceived value from a good or service and its production cost, than marginal competitors within its product market. After that, Sigalas et al. [37] constructed the following two criteria to address a proper definition of FCA: (1) to integrate all latent characteristics of the notion and (2) not to encompass any assessment on its own values or organizational outcomes. They identified FCA as those organizational abilities producing more economic value than the least effective competitors and above the industry average in terms of manipulation of market opportunities, neutralization of competitive threats, and decline of cost. In addition to that, customer satisfaction and optimism, firm reputation, and employee commitment are also acknowledged as essential elements of FCA [18,26,36].

### 2.6. Entrepreneurial Financial Support, Firm Performance, Firm Innovation, and Firm Competitive Advantage

Access to finance through various programs (e.g., formal finance accessed from financial institutions and banks or informal finance accessed from friends, family, and money lenders) has been determined as a positive antecedent of the FiPer of a firm, which is indicated by number of sales [89], higher financial progress, and profitability [90]. FIN provided by the government positively affects stable new ventures' firm performance because enterprises that receive government financial assistance increase their probability of enhanced income, cash flow, and profitability [14,47,91–93]. Peter et al. [94] approved this conclusion by demonstrating that FIN has a significant and positive impact on the FiPer of SMEs by easing financial constraints, reducing risks, and generating economic conditions that promote innovation, entrepreneurial activities, and high profit.

Moreover, firms' access to finance significantly stimulates various facets of BOP including overall growth [95,96], accomplishment of growth and investment opportunities [97], long-term survival of SMEs [98], and the acknowledgement of sustainability challenges [99]. Access to external finance positively influences the BOP, represented by the productivity of labor and organization, of SMEs in both developing and developed nations because financial constraints hinder productivity and represent vital obstacles to efficient entrepreneurial activities [12,100,101]. FIN from governments stimulates organizations by improving their BOP and enlarging their business to achieve higher return on equity, return on assets, and market growth [92]. Kijkasiwat et al. [13] indicated that FIN positively improves BOP, leading to enhanced products and services, production procedures, logistics and delivery, maintenance structure, and organization and administration.

FIN from financial institutions, banks, and other sources positively improves investment in products and processes, leading to the strengthened INO of SMEs in Nigeria [42]. Moreover, FIN through trade credit, asset finance, and overdraft positively improves the INO of new ventures in various nations [44]. The availability of FIN stimulates the INO of MSMEs in India by enabling them to participate actively in innovative activities in which they enforce novel or essential products or processes, novel marketing strategies, or novel organizational techniques in organizational operations, workplace management, and external relationships [102]. FIN demonstrated through the development of financial institutions positively contributes to the INO of enterprises in the EU through mobilizing finance to facilitate firms' patenting activity [43].

FIN positively influences FCA of new ventures since it reconstructs internal procedures and shapes capabilities of new ventures to access resources crucial for developing capabilities, providing FCA [53,103,104]. FIN facilitates FCA because it enables organizations to create returns from distinctions in the valuation employed to a firm between acquisition and divestment and independent of shifts in fundamental outcomes [58,105,106]. Thus, FIN from the government positively stimulates FCA because it helps organizations in reducing numerous costs and assists them in creating particular products and services [47]. Lafuente et al. [107] demonstrated a positive impact of FIN entrenched in an entrepreneurial ecosystem on FCA through the exploitation of resources and competencies, which are adjusted regarding the circumstances of the institutional establishment in which enterprises are operating. We, therefore, propose the following hypotheses:

**H1.** *Entrepreneurial financial support has a positive impact on firm financial performance (H1a), operational performance (H1b), innovation (H1c), and competitive advantage (H1d).*

According to the RBV theory, FCA is acknowledged as the closest driver to performance [40]. FCA allows an enterprise to increase its firm performance compared with its rivals [108,109]. Various scholars have confirmed a significant relationship between FCA and firm performance [92,110,111]. An organization can utilize its FCA to exploit its strengths to assure efficient performance and generate values necessary for sustainable participation in the market, maximizing FiPer and sustaining a high degree of BOP [19,20,49]. Marolt et al. [51] concluded that an SME possessing strong FCA can provide exceptional value to its customers and, thus, it can enhance its sales volume, market share, customer satisfaction, and loyalty. Because an essential objective of firms is to acquire a higher degree of financial outcomes, the obtainment of a continuous FCA is an important factor in achieving this fundamental purpose [112]. Thus, Saeidi et al. [18] proved a positive effect of FCA on the FiPer of manufacturing and consumer product firms in Iran. a positive effect was also confirmed in the context of SMEs functioning as family businesses in Turkey [21]. Jeong and Chung [22] demonstrated that manufacturing SMEs can leverage their FCA to obtain a positive FiPer in the consumer goods sector in Korea. Ofori and Appiah-Nimo [50] suggested that FCA is an essential element of the survival of firms in the hospitality context because of its positive impact on the BOP of hotels in Ghana. Supporting this view, Suandi et al. [52] proposed that organizations with higher levels of FCA can manipulate business

opportunities and neutralize competitor threats, improving the BOP of banks in Indonesia. Thus, we present the following hypothesis:

**H2.** *Firm competitive advantage has a positive impact on firm financial performance (H2a) and operational performance (H2b).*

The RBV theory also suggests that firms can challenge their opponents through generating and improving INO, resulting in FCA and higher levels of firm performance [113]. Despite previous studies concluding mixed results regarding the associations between INO and firm performance [56,57], numerous scholars have explained that INO is a crucial factor benefiting improvements in firm performance. García-Morales et al. [15] and Rita et al. [6] approved the positive impacts of INO on the firm performance of Spanish organizations by stating that an organization adopts an innovative viewpoint to create essential production and technology skills, securing sources for enhanced firm performance. Alipour et al. [45] emphasized that the existence of innovative, skillful, highly qualified, and suitable staff and a particular structure for strategic planning—helping organizations to create efficient approaches and procedures in order to offer novel products and services for their consumers—positively stimulated both the FiPer and BOP of 102 service sector firms in Iran. Moreover, Mai et al. [16] verified the positive influences of INO on the firm performance of tourism and hospitality enterprises in Vietnam. INO—which offers renewal in firms and adaptability in conducting business and enhancing working relationships—enhances the firm performance of companies in Saudi Arabia, promoting their growth and competitiveness in the market [46]. Tripathi and Kalia [17] demonstrated that informational technology enterprises in India can challenge their opponents by enforcing an innovative strategy, i.e., INO, to facilitate FCA and superior firm performance.

INO refers to the development of products, processes, and technologies which help new ventures to enforce novel and effective marketing methods to obtain FCA and, thus, only innovative firms achieve superior firm performance and sustainability in the market in response to business fluctuations [114–116]. Hence, Sulistyo and Ayuni [117] suggested that SMEs, which are constantly seeking innovative approaches to improve their designs and values, are the first to become beneficiaries and, thus, INO stimulates new ventures' sustainable FCA. Azeem et al. [118] demonstrated a positive effect of INO on the FCA of enterprises operating in the textile industry of Pakistan. The improvement and adaptation of novel concepts, behaviors, or procedures in organizational administration strengthens the associations between a firm and its extraneous elements, enabling the firm to enhance its essential capital and generate value for the firm, resulting in FCA [119]. Furthermore, INO is very valuable for developing firms in achieving a sustainable FCA because it encourages the utilization of novel ideas and innovation to create organizational superiority [48,49]. Hence, we propose the following hypothesis:

**H3.** *Firm innovation has a positive impact on firm financial performance (H3a), operational performance (H3b), and competitive advantage (H3c).*

*2.7. Mediating Roles of Firm Competitive Advantage and Firm Innovation*

FCA gained from INO positively affects new ventures' firm performance [19,20,51]. INO allows new ventures to offer more value to customers and maintain their competitive advantage, resulting in better firm performance and profitability [120]. It improves new ventures' firm performance by clarifying and enforcing differentiation and cost leadership strategies [121]. New ventures encounter tough competition when operating in a turbulent market and, thus, they need to develop and enforce innovations that promote their competitiveness in the market, leading to exceptional firm performance [122]. Moreover, the leverage of INO shapes strategic activities necessary for achieving FCA, increasing new ventures' firm performance [123]. Pergelova and Angulo-Ruiz [53] demonstrated that new ventures in the US possessing an innovation-based FCA have higher values of FiPer.

Anwar [114] verified a mediating function of FCA in the positive association between INO and the FiPer of SMEs operating in the emerging market of Pakistan because INO helps firms to obtain a continuous FCA by offering various new methods and generating better profitability and success. Therefore, we propose the following hypothesis:

**H4.** *Firm competitive advantage mediates the effects of firm innovation on firm financial performance (H4a) and operational performance (H4b).*

In both developed and developing nations, FIN has no direct impact on firm performance; instead, internal elements mediate the associations between them [9,53]. Due to the constantly shifting economy and business environment, entrepreneurs and their new ventures have to become innovative to obtain a sustainable FCA [124]. New ventures also exploit the benefits of FIN from the government to embrace new equipment and recruit high-quality staff. It stimulates them to enforce improved INO, ultimately increasing firm performance because of the effectiveness of production and delivery procedures [125,126]. Furthermore, FIN stimulates enterprises in generating and developing unique products and services to achieve their FCA [47]. Moreover, the mediating role of FCA has been broadly researched [47,53,92]. Pergelova and Angulo-Ruiz [53] and Jayeola et al. [9] found that FIN provided by the government does not unveil a direct statistically significant impact on new ventures' firm performance. Nevertheless, FIN demonstrates an indirect impact on firm performance via a mediating contribution of FCA, especially innovation-oriented formation, because FIN enhances original "hard" resources for new ventures, offers them authority among partners, and supports them with a resource "slack" which can be utilized for asset improvement, novel project implement, better market position, and superior firm performance. Songling et al. [92] confirmed that FCA plays a mediating function in the positive associations between FIN and firm performance in an emerging market of Pakistan. Furthermore, Anwar and Li [47] found that FIN enables organizations to reduce numerous costs and create particular products and services, resulting in FCA. These advantages increase profitability and improve the firm performance of SMEs in Pakistan. On the other hand, various articles have emphasized that FIN can be utilized to acquire other internal resources and mechanisms for enterprises including INO [42–44,102]. If an external mechanism (FIN) is accumulated via an internal mechanism (INO), organizations are offered FCA [40], which in turn positively influences firm performance [19,20,51,53]. Therefore, the RBV theory emphasizes that the abilities of an organization to transform available resources into capabilities and resources that are valuable, rare, inimitable, and non-substitutable becomes especially crucial in procedures of gathering and reconfiguring said resources to exploit business opportunities, leading to FCA and superior firm performance [127]. In this circumstance, FIN provides original resource inputs, and then entrepreneurs must develop innovative strategies and procedures that connect to FCA which ultimately mediates the relationships between FIN and new ventures' firm performance. Thus, we present the following hypotheses:

**H5.** *Firm competitive advantage mediates the effects of entrepreneurial financial support on firm financial performance (H5a) and operational performance (H5b);*

**H6.** *Firm innovation mediates the effects of entrepreneurial financial support on firm financial performance (H6a), operational performance (H6b), and competitive advantage (H6c);*

**H7.** *Firm innovation and firm competitive advantage mediate the effects of entrepreneurial financial support on firm financial performance (H7a) and operational performance (H7b).*

Our research's framework is presented in Figure 1.

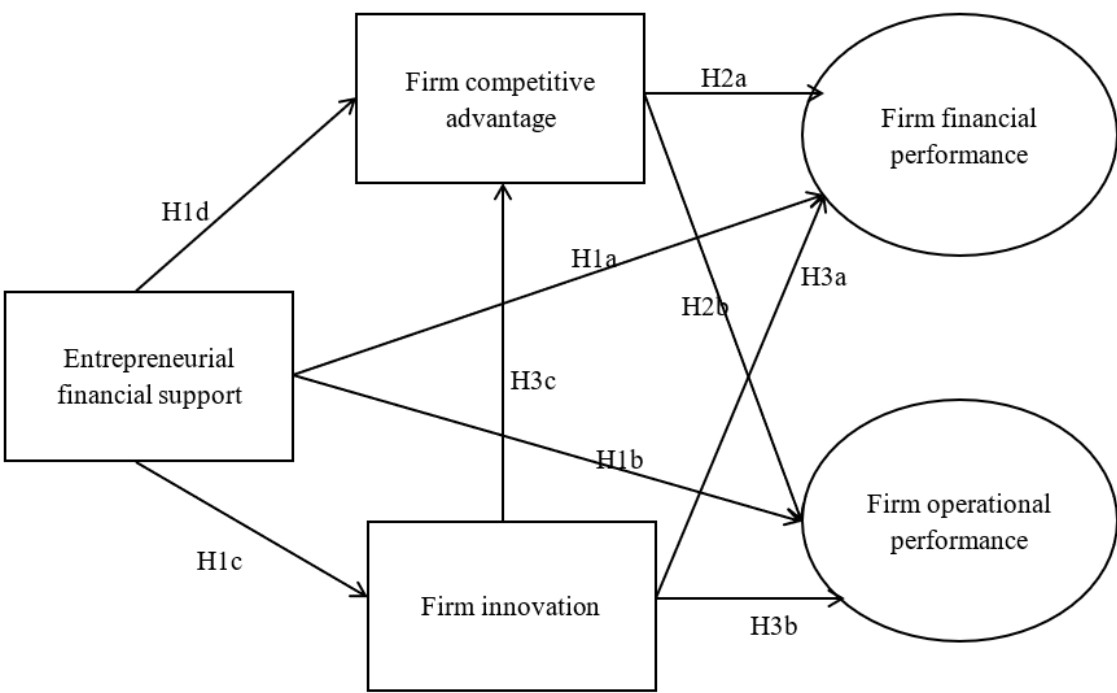

**Figure 1.** Conceptual framework and proposed hypotheses.

## 3. Methodology

### 3.1. Measures

We selected five constructs with 31 indicators to build a questionnaire utilizing extant measurement scales adopted from previous articles: three items used to measure FiPer were adopted from Saeidi et al. [18]; six items for BOP were adopted from Arsezen-Otamis et al. [27]; six items were adopted from Saeidi et al. [18], Sigalas et al. [37], El-Garaihy et al. [36], and Shore et al. [26] to measure FCA; nine items used to measure INO were adopted from García-Morales et al. [15]; and seven items for FIN were adopted from Liguori et al. [79], the World Economic Forum [82], and the Global Entrepreneurship Monitor [83].

After building a conceptual framework and a draft questionnaire following a review of the literature, we performed a pre-test by consulting six experts, namely three researchers in the entrepreneurship field and three entrepreneurs. The pre-test helped us to verify the validity and reliability of our measurement scales and check whether the chosen factors and questionnaire were appropriated in the entrepreneurship context in Vietnam. First, the selected experts assessed five potential factors and their predictive indicators in our conceptual framework. Experts' comments were then collected and considered to adjust our draft questionnaire and, thus, we improved our questionnaire by making necessary revisions regarding their comments including modifications of unclear measurement scales, phrases, and words. Moreover, the content and formation of our questionnaire were reviewed to ensure that they were cohesive and managed for bias. After that, we performed s second pre-test with 12 entrepreneurs to re-evaluate our revised questionnaire and identify the time required to complete it. Finally, we established the final version of our questionnaire, having appropriate measurements which were consistent with extant articles and were appropriate in the Vietnamese context.

The structure of questionnaire included two major sections as follows: the first section measured the respondents' assessment of different measures, namely FiPer, BOP, FCA, INO, and FIN, while the second section obtained demographic data. The final questionnaire consisted of 31 indicators to measure five factors in our conceptual framework, and a "5-point Likert-scale" instrument was utilized to assess each of the constructs, equivalent to "strongly disagree, disagree, neutral, agree and strongly agree respectively" [128]. The

questionnaire was then delivered to entrepreneurs of new ventures operating in the HCMC region, Vietnam.

### 3.2. Sampling Strategies, Sampling Design, and Data Collection

HCMC is a nascent and energetic territory that attracts various new ventures and encourages them to operate [2]. HCMC, as a major urban territory and the largest city in terms of population and area in Vietnam, is the "engine" that drives the remaining regions of Vietnam [129,130]. It is also a major city that generates intimate associations with other regions and nations, contributing the highest percentage of GDP with 15.5 percent in 2022 [3], contributing significantly to the economy and entrepreneurship of Vietnam [4,131]. Thus, among various provinces and cities in Vietnam, the HCMC region was selected as our study area because its characteristics and current conditions fit our research purposes. The target population of this research comprises entrepreneurs who were defined according to three criteria, including (1) entrepreneurs are people who create and manage an enterprise with the ambition of stimulating their firm and performing their leadership abilities to achieve their purposes [132]; (2) they must be associated with new ventures—businesses have been established for no more than 10 years in the HCMC region [133]; and (3) they must voluntarily join this investigation. Entrepreneurs were selected since they could provide a comprehensive viewpoint and rich information associated with research objectives [134]. They are the key criterion in evaluating new ventures [135], being an essential catalyst for INO and a sustainable economy [136]. Hence, they play an important role in the procedure of establishing and developing new ventures [137] because they can determine new business opportunities [138], utilize external resources to provide the necessary financial capital for new ventures, and transform initial ideas into an actual enterprise and, thus, they have deep knowledge about their firms' resources, strategies, and performance [114,139]. In addition to that, the chosen firm age helps capture firms at numerous phases of development, including those in the early, growing, and stabilization phases [140]. This research obtained information related to as-defined organizations through various websites including https://congtymoi.info (accessed on 15 July 2023) and https://thongtindoanhnghiep.co/ (accessed on 15 July 2023). These websites provided a list of new ventures located in Vietnam, which allowed us to access necessary information including company name, established date, company address, industry, telephone number, email of company and entrepreneur, company website, etc. Then, we applied two criteria to select our sample: businesses that (1) have been established for no more than 10 years and (2) are located in the HCMC region. In addition to that, we also obtained detailed information of new ventures operating in the HCMC region by utilizing strong existing relationships with the Planning and Investment Department of the HCMC region. Based on the given information, we selected appropriate entrepreneurs operating within new ventures as our target population and utilized their contact information to conduct our survey. Then, we used convenience sampling and snowball sampling approaches to obtain our data [141,142].

A minimum subjects-to-item ratio of at least 5:1 in EFA was proposed by several scientists as an appropriate sample size for the empirical data collection conducted in this research [143]. The conceptual framework in this research was created with five variables containing 31 items. Therefore, this research required at least 155 cases ($31 \times 5$) of respondents; however, the more respondents, the better [143].

Quantitative data were obtained in two ways. The first way was through conducting online surveys by utilizing Google Forms; questionnaire links were delivered via email and Viber and Zalo applications. The second way was through sending hard-copy questionnaires directly to informants—entrepreneurs in the HCMC region, Vietnam. Considering the 365 respondents comprising the raw data set, there were 50 incomplete questionnaires since a number of entrepreneurs failed to answer all items. The ultimate valid data set comprised 315 finished questionnaires with 215 online respondents and 100 hard-copy respondents from November 2021 to June 2023. This time range illustrated the circumstances

of new ventures during the COVID-19 pandemic and post-COVID-19 pandemic phases. Table 1 displays the profile of respondents.

**Table 1.** Respondents' Profile (N = 315).

| Categories | Items | Frequency (N= 315) | Percentage |
|---|---|---|---|
| Gender | Male | 153 | 48.6 |
| | Female | 162 | 51.4 |
| Age Group | <30 | 97 | 30.8 |
| | 31–40 | 142 | 45.1 |
| | 41–50 | 53 | 16.8 |
| | >50 | 23 | 7.3 |
| Education level | High School | 19 | 6.0 |
| | Vocational | 16 | 5.1 |
| | College | 58 | 18.4 |
| | University | 180 | 57.1 |
| | Postgraduate | 42 | 13.3 |
| Major | Economics | 85 | 27.0 |
| | Social Sciences and Humanities | 26 | 8.3 |
| | Tourism | 17 | 5.4 |
| | Management | 47 | 14.9 |
| Number of staff | Under 10 Employees | 91 | 28.9 |
| | 11–50 Employees | 122 | 38.7 |
| | 51–100 Employees | 42 | 13.3 |
| | Over 100 Employees | 60 | 19.0 |
| Field of company operating | Information Technology | 23 | 7.3 |
| | Transportation | 15 | 4.8 |
| | Agriculture, Forestry, Fishing, and Mining | 12 | 3.8 |
| | Real Estate Activities | 51 | 16.2 |
| | Retail and Distributive Trade | 27 | 8.6 |
| | Service Activities/Tourism | 31 | 9.8 |
| | Manufacturing | 50 | 15.9 |
| | Others | 106 | 33.7 |
| Total annual revenue | Under 10 Billion | 184 | 58.4 |
| | 11-100 Billion | 85 | 27.0 |
| | Over 100 Billion | 46 | 14.6 |

*3.3. Analysis*

This research investigated the relationships among FIN, INO, FCA, FiPer, and BOP. It also explored the mediating roles of INO and FCA in the relationships between FIN and FiPer and BOP using a pragmatic quantitative approach [143]; this research mainly used quantitative approaches and the partial least squares structural equation modeling (PLS-SEM) technique to conduct an empirical test of our research hypotheses [143]. We applied PLS-SEM because this study constituted exploratory research and the indicators were considered reflective measurements [143]. It is a statistical tool with great power in identifying whether proposed relationships are significant [144]. Moreover, PLS-SEM provides an opportunity to analyze the data in greater detail. In this study, the research model was complex and included various constructs, including FIN, FCA, INO, FiPer, and BOP, having 31 indicators to measure five factors in our conceptual framework and, thus, PLS-SEM was used to predict the relationships among these constructs [144,145]. PLS-SEM addresses small sample sizes and also works very well with large sample sizes [144]. PLS-

SEM was utilized to examine the variance among constructs, and it also was appropriate for the investigation of complicated constructs. PLS-SEM concentrates on maximizing the variance of endogenous latent variables demonstrated through exogenous variables in reverse to attempt to reflect the experimental covariance matrix [146,147]. We utilized PLS-SEM by using Smart-PLS software version 3.0 to assess our research model [143] in order to process the 315 cases. With 2000 replications, non-parametric bootstrapping was assessed [148]. We used a two-step technique to analyze the obtained data [143]. In the first step, the 315 valid datapoints were checked for reliability and validity of measurement scales of the outer model by using composite reliability (CR), average variance extracted (AVE), and Cronbach's alpha. In the second step, we concentrated on determining the potential associations among constructs, the structural model evaluated using pertinent results of measurements in our research model, and the significance and impacts of path coefficients. Hair et al. [144] proposed that "PLS is used for prediction-oriented research that aims to maximize the explained variance of dependent variables and can be used if less rigid theoretical backgrounds are available". To be more specific, this research has two sub-models including an inner model and an outer model [149].

Focusing on the inner model, it explains the relationships between exogenous variables (FIN, INO, and FCA) and endogenous latent variables (FiPer and BOP); combined with the outer model, it can explain the associations between latent variables and their observed indicators. The structural equation model was used to investigate our research hypotheses through assessing the path coefficient sizes and significance of the inner model (β), utilizing the non-parametric bootstrapping technique [148].

## 4. Data Analysis and Results

### 4.1. Demographic Characteristics of Samples

Demographic analysis was conducted through SPSS 20 to collect information from respondents. The final valid data set includes 315 respondents and the profiles of the respondents include seven categories related to demographic statistics including gender, age group, education level, major, number of staff, field of company operation, and total annual revenue (see Table 1).

### 4.2. Measurement Model Results

In the first stage, we determined the convergent validity and consistency reliability for each indicator and applied CR and AVE to examine them. CR was used to determine internal reliability [150] and AVE was exploited to evaluate the convergent validity [151]. The minimum value for CR was 0.7 and 0.5 for AVE [151,152]. Table 2 shows the results of the CR of all constructs. Table 2 demonstrates that CR values ranged from 0.925 to 0.956, which is in line with Hair et al. [143] who proposed that values of CR equal to 0.6 or higher are acceptable. The AVE values ranged from 0.671 to 0.867 for each factor, which is in line with Hair et al. [144] who concluded that the AVE values equal to 0.5 or higher are acceptable. Thus, this result illustrated that all constructs express the model's high degrees of internal consistency, reliability, and convergent validity. Cronbach's alpha was utilized as the primary method to evaluate internal consistency and reliability. Cronbach's alpha values greater than 0.7 indicated suitable reliability of the measured constructs [144]. The Cronbach's alpha values in our research ranged from 0.902 to 0.948. Hair et al. [143] also demonstrated that values of factor loading that do not exceed 0.60 should be eliminated. However, in our research, there were no eliminated indicators because all the indicators of the five evaluated constructs possessed factor loadings that were higher than 0.60 (see Table 2).

**Table 2.** Properties of the constructs.

| Constructs and Indicators | Factor Loading |
|---|---|
| Financial Performance: FiPer (Cronbach's Alpha = 0.923, CR = 0.951, AVE = 0.867) | |
| FiPer1: In comparison with competitors' return on sales increases | 0.934 |
| FiPer2: In comparison with competitors' return on assets increases | 0.929 |
| FiPer3: In comparison with competitors' return on equity increases | 0.930 |
| Firm Operational Performance: BOP (Cronbach's Alpha = 0.911, CR = 0.931, AVE = 0.692) | |
| BOP1: Our firm can find credits easily when needed | 0.763 |
| BOP2: The customers are satisfied with the firm | 0.844 |
| BOP3: We present enough new products/menus/services for the customers | 0.799 |
| BOP4: Our firm has a competitive advantage | 0.859 |
| BOP5: We get the worth of our money, labor, and time we spent for the firm | 0.864 |
| BOP6: Our company is successful in general | 0.857 |
| Firm Competitive Advantage: FCA (Cronbach's Alpha = 0.902 CR = 0.925, AVE = 0.671) | |
| FCA1: Exploit all market opportunities that have been presented to your industry | 0.828 |
| FCA2: Neutralize all competitive threats from rival firms in your industry | 0.843 |
| FCA3: Reduction of total expenses at a higher rate than competitors | 0.811 |
| FCA4: The employee appears to highly committed to the organization | 0.803 |
| FCA5: Customers are satisfied with our firm's products and services | 0.815 |
| FCA6: Customers are optimistic about long-term future of this corporation | 0.815 |
| Firm Innovation: INO (Cronbach's Alpha = 0.949, CR = 0.956, AVE = 0.709) | |
| INO1: Organization's emphasis on developing new products or services | 0.854 |
| INO2: Rate of introduction of new products or services into the market | 0.842 |
| INO3: Organization's spending on new product or service development activities | 0.858 |
| INO4: Number of new products or services added by the organization and already on the market | 0.841 |
| INO5: Number of new products or services that the organization has introduced for the first time to the market | 0.826 |
| INO6: Investment in developing proprietary technologies | 0.800 |
| INO7: Emphasis on creating proprietary technologies | 0.861 |
| INO8: Organization's emphasis on technological innovation | 0.844 |
| INO9: Organization's emphasis on pioneering technological developments in its industry | 0.851 |
| Entrepreneurial financial support: FIN (Cronbach's Alpha = 0.930, CR = 0.944, AVE = 0.707) | |
| FIN1: There are local individual investors in my community who are willing to financially support entrepreneurial ventures | 0.748 |
| FIN2: New and growing firms have opportunities to raise capital from friends and family | 0.788 |
| FIN3: Bankers in my community work hard to help entrepreneurs obtain financing | 0.870 |

**Table 2.** *Cont.*

| Constructs and Indicators | Factor Loading |
|---|---|
| FIN4: Financing for entrepreneurship is available in my local community | 0.890 |
| FIN5: Information on what funding programs are available for entrepreneurs is easily accessible | 0.855 |
| FIN6: My community has a sufficient number of banks who are willing to lend to entrepreneurs | 0.870 |
| FIN7: There are sufficient government subsidies available for new and growing firms | 0.857 |

CR: composite reliability; AVE: average variance extracted.

*4.3. Discriminant Validity*

To check the discriminant validity, this research implemented the ratio proposed by Fornell and Larcker [151] to determine the square root of AVE values: each latent variable should be higher than the correlations among constructs. It can be leveraged to estimate discriminant validity in the circumstance where the square root of an AVE value is higher than other correlation values among latent variables. In addition to that, other well-identified latent variables assure discriminant validity according to Hair et al. [143] who recommended that "an indicator's loadings should be higher than all of its cross loadings". Table 3 illustrates the results of discriminant validity, which supported all constructs, ranging from 0.819 to 0.931.

**Table 3.** Discriminant validity of Fornell and Larcker criteria results.

| | FIN | FiPer | FCA | INO | BOP |
|---|---|---|---|---|---|
| FIN | **0.841** | | | | |
| FiPer | 0.584 | **0.931** | | | |
| FCA | 0.662 | 0.625 | **0.819** | | |
| INO | 0.696 | 0.588 | 0.718 | **0.842** | |
| BOP | 0.655 | 0.690 | 0.783 | 0.720 | **0.832** |

Bold values represent the square root of AVE values.

*4.4. Assessment of the Structural Model*

4.4.1. Testing Multicollinearity

In order to test the existence of multicollinearity, the variance inflation factor (VIF) was used. According to Hair et al. [144], a VIF value lower than 4 is permissible. If the VIF value is higher than 5, multicollinearity generates issues in predictor variables. According to the collinearity statistics in our research, VIF values ranged from 1.846 to 4.067, indicating that multicollinearity was not an issue in our data. The measure outcomes of our conceptual framework were acceptable. Furthermore, an explanation of target endogenous variables' (FiPer, BOP, FCA, and INO) variance was leveraged to examine the sufficiency of our predictive model.

4.4.2. Testing Predictive Power of Structural Model

To predict the power of our model, the estimated $R^2$ weight of endogenous constructs was valued. Our results point out that the coefficient of determination ($R^2$) was 0.455 for FiPer. This explained that seven latent variables (FCA, INO, FIN) in the structural model can be illustrated as substantial, which explained 45.5 percent of the variance in FiPer. The $R^2$ weight of BOP was 0.674, indicating that five latent variables (FCA, INO, FIN) in the structural model can be demonstrated as substantial, which explained 67.4 percent of the variance in BOP. The $R^2$ weight of FCA was 0.567, indicating that five latent variables (INO, FIN) in the structural model can be illustrated as substantial, which explained 56.7 percent of the variance in FCA. The $R^2$ weight of INO was 0.485, indicating that five latent variables (FIN) in the structural model can be demonstrated as substantial, which

explained 48.5 percent of the variance in INO. In this research, the results of $R^2$ were substantial and moderate regarding the recommendation of Hair et al. [143].

### 4.4.3. Testing Predictive Relevance

We utilized blindfolding to measure the predictive relevance, and an instrument was applied to assess the inner model. The value of $Q^2$ was higher than zero and, thus, exogenous constructs were of predictive relevance for endogenous variables with acceptable model fit [143]. In this research, the average cross-validated redundancy values reached 0.388 for FiPer, 0.459 for BOP, 0.373 for FCA, and 0.338 for INO, all higher than zero. Hence, it can be concluded that a high predictive relevance for FiPer, FCA, INO, and BOP was demonstrated, exhibiting an adequate model fit according to Hair et al. [143]. Therefore, there was predictive relevance for FiPer, FCA, INO, and BOP in the research model.

Figure 2 displays the results of hypothesis testing; this research examined the structural model at a 95% level of confidence following "non-parametric bootstrapping" with 2000 replications [143,144].

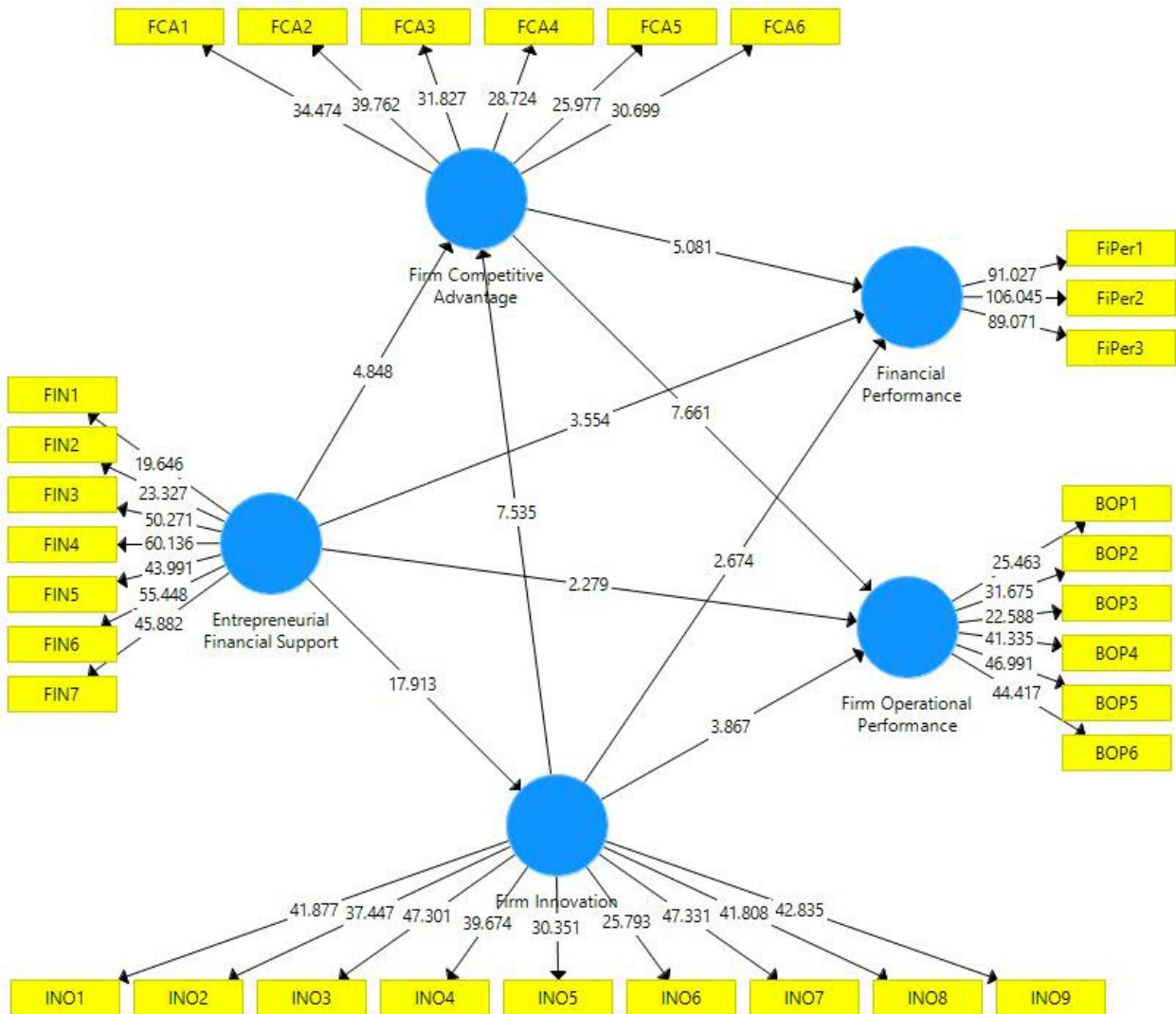

**Figure 2.** Results of structural equation model.

### 4.5. Hypotheses Testing: Direct Effects

Figure 2 illustrates the structural model, which is a result of our PLS analysis. Hypothesis 1 (H1a, H1c, H1c, H1d) was totally supported: FIN has a positive relationship with FiPer ($\beta = 0.231$, $p = 0.000$), BOP ($\beta = 0.137$, $p = 0.023$), INO ($\beta = 0.696$, $p = 0.000$),

accounting for the highest impact, and FCA ($\beta = 0.313$, $p = 0.000$). It was indicated that a single-unit change in FIN caused increases of 0.231 units in FiPer, 0.137 units in BOP, 0.696 units in INO, and 0.313 units in FCA (see Table 4).

**Table 4.** Summary results of direct effects.

| Hypothesis | Direct Effects | Path Coefficients | T-Values | *p*-Values | Decisions |
|---|---|---|---|---|---|
| H1a | FIN → FiPer | 0.231 | 3.554 | 0.000 | Supported |
| H1b | FIN → BOP | 0.137 | 2.279 | 0.023 | Supported |
| H1c | FIN → INO | 0.696 | | 0.000 | Supported |
| H1d | FIN → FCA | 0.313 | 17.91 | 0.000 | Supported |
| H2a | FCA → FiPer | 0.341 | 4.848 | 0.000 | Supported |
| H2b | FCA → BOP | 0.505 | 5.081 | 0.000 | Supported |
| H3a | INO → FiPer | 0.183 | 7.661 | 0.008 | Supported |
| H3b | INO → BOP | 0.262 | 2.674 | 0.000 | Supported |
| H3c | INO → FCA | 0.500 | 3.867 | 0.000 | Supported |

Hypothesis 2 (H2a, H2b) was totally supported: FCA has a positive relationship with FiPer ($\beta = 0.341$, $p = 0.000$), and BOP accounted for the highest impact ($\beta = 0.505$, $p = 0.000$). It was indicated that a single-unit change in FCA caused increases of 0.341 units in FiPer and 0.505 units in BOP (see Table 4).

Hypothesis 3 (H3a, H3c, H3c) was totally supported: INO has a positive relationship with FiPer ($\beta = 0.183$, $p = 0.008$), BOP ($\beta = 0.262$, $p = 0.023$), and FCA accounted for the highest impact ($\beta = 0.500$, $p = 0.000$). It was indicated that a single-unit change in INO caused increases of 0.183 units in FiPer, 0.262 units in BOP, and 0.500 units in FCA (see Table 4).

*4.6. Mediation Analysis*

To consider the mediating effects of INO and FCA, from the specific indirect effect results, it could be concluded that because of the significant influences of FIN on both INO and FCA that were directly related to FiPer and BOP, these exogenous constructs had indirect effects on FiPer and BOP via the mediations of both INO and FCA. Table 5 shows the results of mediating variables. All the hypotheses (H4a, H4b, H5a, H5b, H6a, H6b, H6c, H7a, and H7b) were totally supported.

**Table 5.** Summary Results of Indirect Effects.

| Hypothesis | Indirect Effects | Path Coefficients | T-Values | *p*-Values | Decisions |
|---|---|---|---|---|---|
| H4a | INO → FCA → FiPer | 0.170 *** | 3.348 | 0.001 | Supported |
| H4b | INO → FCA → BOP | 0.252 *** | 5.060 | 0.000 | Supported |
| H5a | FIN → FCA → FiPer | 0.107 ** | 4.464 | 0.000 | Supported |
| H5b | FIN → FCA → BOP | 0.158 *** | 4.154 | 0.000 | Supported |
| H6a | FIN → INO → FiPer | 0.127 * | 2.599 | 0.009 | Supported |
| H6b | FIN → INO → BOP | 0.182 *** | 3.631 | 0.000 | Supported |
| H6c | FIN → INO → FCA | 0.348 *** | 6.569 | 0.000 | Supported |
| H7a | FIN → INO → FCA → FiPer | 0.119 *** | 4.143 | 0.000 | Supported |
| H7b | FIN → INO → FCA → BOP | 0.176 *** | 4.747 | 0.000 | Supported |

\* $p < 0.05$, \*\* $p < 0.01$, \*\*\* $p < 0.000$ (one tail).

## 5. Discussion

This research succeeded in conducting empirical research to determine the role of FIN in enhancing the FCA, INO, and firm performance of new ventures in the HCMC region, Vietnam. In addition, this research succeeded in investigating the mediation roles of FCA and INO to examine the causal relationships between the FIN and firm performance of new ventures. Besides that, this research utilized the RBV theory to analyze

the associations between FIN and firm performance through the mediating roles of FCA and INO. Our results are provided according to an analysis of 315 respondents gathered from entrepreneurs operation within new ventures in the HCMC region, Vietnam.

The first research question in this research was clarified as follows: To what extent does entrepreneurial financial support directly and indirectly affect new ventures' firm performance, competitive advantage, and innovation? This research confirmed that FIN has significant and direct positive influences on indicators of firm performance of new ventures including FiPer (H1a) and BOP (H1b) as well as INO (H1c) and FCA (H1d). Thus, our findings supported the RBV theory of Barney [40] who proposed that the sustainable FCA and outcomes of a firm are the result of the possession of unique organizational resources that are valuable, rare, inimitable, and non-substitutable. Specifically, this research demonstrated positive effects of FIN on the FiPer and BOP of new ventures. Hence, our results are in conflict with those of previous studies [9,53–55] that claim that FIN does not have a direct statistically significant effect or even has negative impact on new ventures' firm performance. Regarding the influence of FIN on the FiPer of new ventures, our results were in line with extant studies that pointed out that FIN positively improves the FiPer of a firm [94] reflected through increased number of sales [89,90], cash flow, profitability, and incomes of enterprises [14,47,91–94]. Regarding the positive influence of FIN on the BOP of new ventures, our results were in line with extant investigations that claimed that FIN helps firms to achieve overall growth and long-term survival [95–99], having positive impacts on their BOP [14,76]. In addition, by demonstrating the positive effect of FIN on the BOP of new ventures in Vietnam, this research favored the conclusions of Chowdhury et al. [12], Boermans and Willebrands [100], and Ferrando and Ruggieri [101] who suggested that FIN has a positive impact on the labor productivity and organizational productivity of new ventures in developing countries. Our results demonstrated that FIN had the strongest direct effect on INO in our model, which was in line with extant research [42–44] claiming that FIN positively stimulates the INO of SMEs by enhancing investment in products and processes and mobilizing finance to facilitate firms' patenting activity. In addition to that, our results strongly flavored those of Kaur et al. [102] who proposed that FIN has a positive influence on INO of MSMEs in a similar context as that of Vietnam: India—an Asian country. Regarding the positive effect of FIN on FCA, our findings strongly supported the results of Pergelova and Angulo-Ruiz [53], Hansen et al. [104], Doh and Kim [103], and Lafuente et al. [107] who claimed that FIN positively influences the FCA of new ventures because it helps new ventures to access crucial resources to develop necessary capabilities to obtain a sustainable and competitive position. Similarly, FIN enables new ventures to achieve FCA since it stimulates enterprises in generating returns from differentiation in the valuation between acquisition and divestment and independent of changes in their performance [58,105,106]. Our results are similar to the findings of Anwar and Li [47] who suggested that FIN helps new ventures in reducing numerous costs and producing unique products and services.

Therefore, this research surpassed previous studies by considering a comprehensive viewpoint of FIN that included extensive external financial resources of new ventures including local investors, local communities, friends and family, banks, bankers, funding programs, and government subsidies instead of only focusing on FIN provided by the government [14,47,53,89,91–94,103,104] or other non-government support institutions [91] or access to finance though investment climate, banks, and non-bank financial institutions [13,42–44,58,95–102]. Thus, this research contributed to the entrepreneurship literature by creating and validating a comprehensive measurement to investigate FIN systems that can be leveraged in future studies to enhance knowledge in this research field. Besides that, by confirming that FIN has positive effects on new ventures' firm performance, our findings solved extant debates surrounding FIN–firm performance links [9,53–55] that present differing opinions on whether FIN has a direct impact or no direct influence or even a negative impact on new ventures' firm performance. In addition, through confirming the positive effects of FIN on extensive indicators of new ventures' firm performance including

FiPer and BOP, this research improves upon existing studies that investigated the influences of FIN on FiPer [14,47,89–94] and BOP [12,13,92,95–101] separately. Furthermore, it demonstrated the positive impacts of FIN on INO and FCA, offering thorough outcomes of FIN in a single investigation to expand the entrepreneurship literature.

Moreover, we demonstrated positive influences of FCA on new ventures' FiPer and BOP and, thus, supported H2a and H2b. Therefore, this research also strengthened the RBV theory of Barney [40], which claimed that FCA is considered the closest driver to firm performance. Our findings favored those of Yaskun et al. [49], Mukhsin and Suryanto [19], and Astuti et al. [20] who concluded that FCA positively affects FiPer and BOP of SMEs. In particular, our findings were in line with extant investigations that clarified the influences of FCA on FiPer [18,21,22,112] and BOP [50–52] in various industries and nations.

Furthermore, we also concluded that INO positively contributes to new ventures' FiPer, BOP, and FCA and, thus, supported H3a, H3b, and H3c. Thus, this research also enhanced the RBV theory of Barney [40] by analyzing and confirming INO as an internal mechanism that stimulates new ventures in enhancing their sustainable FCA [73] and firm performance [69,74]. Focusing on the impacts of INO on new ventures' FiPer and BOP, unlike Rosenbusch et al. [56] and Li and Atuahene-Gima [57] who presented mixed results of the associations between INO and firm performance, our results supported previous findings of Rita et al. [6], García-Morales et al. [15], Tripathi and Kalia [17], and Defalla and Choong [46] who proposed positive impacts of INO on firm performance because enterprises can leverage an innovative perspective to produce crucial products and technology skills to develop their growth and competitiveness, resulting in high performance. Mai et al. [16] concluded a similar relationship, also in Vietnam. Similarly, Alipour et al. [45] claimed that the existence of INO helps firms to generate efficient approaches and procedures that provide novel products and services to their consumers, positively stimulating both FiPer and BOP. Another important result was that INO positively affects FCA, supporting previous studies [49,114–118] that suggest that only innovative enterprises sustain stability in the market in response to business fluctuations and, thus, they indicate the positive influence of INO on FCA. Specifically, our results strongly flavored those of Alfawaire and Atan [119] who suggested that INO allows enterprises to improve their valuable capital, create value, and develop their FCA. Similarly, Banmairuroy et al. [48] found that INO helps firms to create a sustainable FCA since it facilitates the operation of new ideas and innovation to create organizational superiority. Considering various forms of innovation such as products, processes, administration, technology, exploration, economic exploitation, and firm innovation, this research selected and investigated INO, which encompasses the organizational innovation of new ventures thoroughly, to depict the comprehensive internal capabilities of innovative new ventures. Moreover, instead of examining the effects of INO on the overall firm performance of new ventures [6,15–17,46], this research expands upon previous studies by confirming the positive effects of FIN on two distinct indicators of new ventures' firm performance, namely FiPer and BOP, providing explanations of these relationships to the existing literature. Furthermore, by confirming the positive effects of INO on new ventures' firm performance, this research resolved current debates in INO–firm performance links [56,57] that illustrate mixed results.

For the second research question, this research focused on evaluating the significance of mediating variables of firm innovation and competitive advantage in determining the relationships between entrepreneurial financial support and the firm performance of new ventures. Our results completely supported H4 (H4a, H4b), which favored previous findings of Anwar [114], Teece [120], and Pellikka and Malinen [122] who proposed that FCA is a significant mediating variable in the positive relationships between INO and firm performance because INO enables firms to achieve sustainable FCA by developing and adapting numerous novel concepts and methods. Hence, these firms offer more value to customers and remain competitive, leading to better profitability, performance, and success. In support of this view, INO increases firm performance by forming and

implementing differentiation, cost leadership, and other strategic strategies needed to achieve FCA [121,123].

Another essential finding was that FCA positively mediates the relationships between FIN and new ventures' FiPer and BOP (H5a, H5b). This finding was in line with existing articles [53,92] that found mediating roles of FCA in associations between FIN and firm performance because FIN enhances initial "hard" resources for firms, which can be leveraged for asset improvement, new project implementation, better market positions, and superior firm performance. Similarly, Anwar and Li [47] concluded that FIN enables SMEs to reduce numerous costs and stimulates them to create particular products and services, resulting in increased profitability and improved firm performance.

Our results also demonstrated mediating roles of INO in associations between FIN and new ventures' FiPer, BOP, and FCA (H6a, H6b, H6c); FIN has the strongest indirect effect on FCA via INO. Therefore, our findings were in line with existing papers [125,126] stating that new ventures utilize FIN to adopt new equipment and recruit high-quality staff in order to develop INO, ultimately leading to increased firm performance because of the efficiency of production and delivery processes. Furthermore, our findings supported previous studies [42–44,102] that proposed that FIN can be utilized to acquire an internal mechanism like INO. In the case where an external mechanism (FIN) is combined with an internal mechanism (INO), firms achieve FCA [40].

Thus, combining the verified mediating role of INO with the positive relationships between FIN and new ventures' firm performance, this study expanded upon previous investigations, which debated whether FIN has a direct impact [12–14,47,89–101], no direct influence [9,53], or even a negative impact on new ventures' firm performance [54,55], by concluding that FIN has both positive direct and indirect effects on indicators of new ventures' firm performance including FiPer and BOP via an internal mechanism (INO), offering an explanation of more holistic relationships between them.

Finally, the mediating effects of FCA and INO in the relationships between FIN and new ventures' firm performance were confirmed (H7a, H7b). These findings expanded upon the RBV theory because they indicated that new ventures' competencies to transform available resources into valuable, rare, inimitable, and non-substitutable capabilities and resources enable those ventures to exploit business opportunities, resulting in FCA and, ultimately, superior performance [127]. In a supporting point of view, previous studies [19,20,51,53] claimed that an FCA obtained from external and internal mechanisms positively influences firm performance and, thus, they illustrated the mediating impacts of FCA and INO on the associations between FIN and firm performance. Therefore, this research was differentiated from previous studies and possessed novelty because it was a pioneering study providing a comprehensive picture of relationships between FIN and new ventures' firm performance through mediating roles of FCA and INO, which has not been fully examined in the literature. It was proposed that FIN captured by new ventures is utilized to develop internal competing resources and capabilities (INO) to first stimulate new ventures in obtaining sustainable FCA before resulting in improved firm performance in terms of FiPer and BOP, offering complicated mechanisms underlying the positive relationships between FIN and new ventures' firm performance, in opposition to uncomplicated mediations of FCA and INO analyzed in existing studies, in order to fully understand RBV theory.

*5.1. Practical Implications*

This research offers entrepreneurs various practical mechanisms to enhance their new ventures' firm performance during the COVID-19 pandemic and post-COVID-19 pandemic phases. Moreover, it also provides governors and other stakeholders methods to improve FIN systems to enhance new ventures' firm performance, potentially leading to successful entrepreneurship in a specific region.

Firstly, FIN was the major factor that improved new ventures' FiPer and BOP during the COVID-19 pandemic. Hence, governors and other stakeholders of a specific region

should implement renewed commitment to stimulate the overall development of new ventures in Vietnam. They must be aware of developing an appropriate FIN system in the post-COVID-19 pandemic era. Therefore, they should take the initiative to invest time and money into creating and promoting a system that provides efficient financial assistance to new ventures operating in their region. This research offers a comprehensive viewpoint of FIN including local investors, local community, friends and family, banks, bankers, funding programs, and government subsidies. Hence, there are various useful strategies that can be pursued to improve FIN systems. Stakeholders should eliminate all unnecessary bureaucratic barriers to accessing financial support, enhance financial assistance programs, stimulate funding in their local community, and reduce interest rates of loans and taxes. Besides that, they should also offer strategic venture capital, government subsidies, funding programs, and other types of financial organizations for new ventures to provide funds at an adequate accessible rate to all crucial business industries. Furthermore, the government should use capital from the state budget to research and develop sustainable entrepreneurship. Moreover, we encourage stakeholders to modify and develop better policies and procedures that stimulate local investors, banks, and bankers to provide appropriate capital to new ventures. They can also promote funding programs through both traditional and modern channels to ensure that all entrepreneurs can approach and access information on said programs easily. We also encourage the Small and Medium Enterprise Development Authority and other responsible institutions to support new ventures financially. Finally, it is suggested for entrepreneurs to create and strengthen relationships with government, political, and financial institutions and other organizations and individuals because they can raise capital efficiently from extant relationships such as those of friends, family, and other individuals and organizations. Hence, by having strong FIN systems, entrepreneurs and their new ventures can access and obtain adequate external resources—finance—in an effective way and, thus, have the sufficient financial capital for their ventures to operate and survive, resulting in exceptional firm performance and success.

Secondly, our findings also offer entrepreneurs and their new ventures a proper viewpoint of INO and new insight into how to build and utilize INO to enhance new ventures' FiPer and BOP because INO is a significant driver that contributed positively to firm performance during the COVID-19 pandemic. Hence, this research provides entrepreneurs and their new ventures with numerous valuable strategies that can be used to enhance INO in the post-COVID-19 pandemic era. It is necessary for entrepreneurs to promote an environment that facilitates innovation within their new ventures to make INO a core value and, thus, ensure INO among all organizational members. Entrepreneurs and their new ventures must evaluate all of their manufacturing and technical resources that stimulate their performance and success. Moreover, they should emphasize and invest in hiring employees with different perspectives, developing new products and services, and creating feedback processes and reward systems. This will lead to a high rate and high number of new products and services being introduced into the market and, thus, offering noteworthy value to their customers. In addition, entrepreneurs should also ensure that said new products and services possess unique and distinct values compared with those of their opponents. Furthermore, entrepreneurs are required to invest in developing proprietary technologies, adopting novel technologies, offering training to promote technological innovation, and enabling innovative products and processes.

Thirdly, FCA was another crucial factor that enhanced new ventures' firm performance during the COVID-19 pandemic. Thus, this research states that new ventures should strengthen themselves to achieve a strong competitive position in order to improve their firm performance in the post-COVID-19 pandemic era. Therefore, entrepreneurs and their new ventures are required to exploit all business opportunities and neutralize all competitive threats by identifying their unique strengths and weaknesses. They should analyze their business activities and their opponents to ensure benefits of cost saving and diversification in more efficient ways compared to their opponents. Besides that,

entrepreneurs should create a supportive climate in which employees are encouraged to commit to their firms. They should also concentrate on building an efficient quality control system, customer services, and strong brand awareness. These strategies help new ventures to create valuable, rare, inimitable, and non-substitutable capabilities and resources that offer more exceptional and valuable products and services to their customers, ultimately producing exceptional firm performance that exceeds that of their competitors.

Finally, this is a pioneering study investigating and confirming the relationships between FIN and firm performance through the mediating roles of FCA and INO. Our findings show that a new ventures' combination of an external mechanism—FIN—and an internal mechanism—INO—secures resources in order to achieve sustainable FCA, resulting in exceptional firm performance. In modern business, because sustainability is the most crucial element for new ventures in enhancing firm performance, governors and other stakeholders should create a beneficial FIN system, while entrepreneurs should leverage said system to obtain sufficient financial capital for their new ventures to develop and implement products, processes, and technological innovations. Hence, by efficiently using a combination of external and internal mechanisms, new ventures should develop their own valuable, rare, inimitable, and non-substitutable resources and capabilities to achieve sustainable FCA that is considerate of communities, governments, and the natural environment. In order to secure a competitive advantage, new ventures must develop strategies and business models that are agile, and they should focus on exploiting eco-friendly products and services by using renewable energy, local sources, and alternative materials. This transformation is considered as a sustainable source through which new ventures to become successful by generating higher values of FiPer and BOP compared to those of their rivals. Thus, it ultimately influences the long-term survival and sustainability of new ventures, national economic growth, and sustainable nationwide progress. Because sustainable development includes three primary factors, namely economic development, environmental protection, and social well-being elements—which create opportunities for competitive advantage, improved performance of new ventures, innovation, and national economic growth—it transforms into sustainable entrepreneurship and national sustainable economic recovery by inciting a revolutionary change in the way we approach the crucial facets of sustainable development.

*5.2. Limitations and Future Research*

This research has various implications in both theory and practice, but it is not free of limitations, which should be addressed in forthcoming studies. Firstly, we only conducted this research in the HCMC region of Vietnam—a developing country—and, thus, it was confined to the HCMC region and might not be seen as a good representation of the entirety of Vietnam and the world, especially developed economies. Thus, more evidence should be obtained from other contexts and developed nations to achieve more valuable understandings. Secondly, this study only focused on SMEs; our findings reported that the enterprises with lower than 100 employees accounted for 80 percent, so our results were not generalizable to all enterprises in terms of SMEs and large enterprises of Vietnam. Future studies should expand their data sets to collect data from both SMEs and large enterprises with sufficiently large sample sizes which may yield meaningful results. Thirdly, this study used PLS-SEM to predict the influence of FIN on the INO, FCA, and firm performance of new ventures. Therefore, future research can conduct mixed-methods research or post hoc analysis and add new factors to build a more comprehensive entrepreneurial ecosystem in other research areas. In this regard, a study of the factors "policy, culture, supports, human capital, markets, R&D transfer, and networks" can help in new crises because of their roles in the process of new ventures' success because said factors can influence the variables proposed in our research framework [59]. Hence, we believe that forthcoming research should employ a wider scope in the nomological network of the entrepreneurial ecosystem [59] to examine appropriate antecedents and mediating variables, associating with related theories to obtain a deeper knowledge of predictors of new ventures' firm

performance. Therefore, an analysis of the effect of these factors that can manage crises and motivate elements for enterprises in other fields is an interesting direction for future studies. Finally, this research focused only on INO of new ventures. Because innovation possesses other specific facets including product or process innovation [153], administrative or technological innovation [154], and exploratory or exploitative innovation [155], it is crucial to examine how other types of organizational innovation can influence the relationships between FIN and new ventures' firm performance.

## 6. Conclusions

By using a database of 315 entrepreneurs of new ventures operating in the HCMC region, this research investigated the impacts of FIN on new ventures' FiPer and BOP via the mediating roles of FCA and INO. A structured questionnaire was utilized to collect data from entrepreneurs. After using PLS-SEM, this research provided various findings that contributed significantly to the extant literature. First, this research generated and validated a comprehensive measurement of FIN, including extensive external financial sources for new ventures by utilizing the most common measurement scales demonstrated in extant systematic literature reviews in order to enhance the causal relationships of FIN, strengthening the sustainable entrepreneurship literature. Second, our results demonstrated that FIN has significant, direct, and positive impacts on new ventures' FiPer, BOP, INO, and FCA. Hence, this research solved current debates whether FIN has a direct impact, no direct influence, or even a negative impact on new ventures' firm performance, while offering extensive outcomes of FIN to stimulate the awareness of other scholars about the importance of FIN to expand upon the entrepreneurship literature. Third, this research also proved that FCA and INO positively influenced the firm performance of new ventures, resolving ongoing debates in INO–firm performance associations, concurrently providing a broad range of predictors of firm performance that strengthened the antecedents of new ventures' firm performance. Finally, this research was a pioneering study examining and confirming mediating functions of FCA and INO in relationships between FIN and the firm performance of new ventures, which has not been fully examined in the literature. Thus, these confirmed mediating functions demonstrated our research's novelty and differentiation. Thus, this research enhanced the RBV theory in the entrepreneurship context by confirming that new ventures can achieve stronger FCA, resulting in higher firm performance, by utilizing FIN systems as an external mechanism and developing INO as an internal mechanism.

**Author Contributions:** K.N.M. created the research framework and, together with Q.H.T., was engaged in data collection and analysis; Q.H.T. wrote the article; K.N.M. checked the final version of our manuscript and contributed to the submission. All authors have read and agreed to the published version of the manuscript.

**Funding:** This research received no external funding.

**Institutional Review Board Statement:** Not applicable.

**Informed Consent Statement:** Not applicable.

**Data Availability Statement:** The data presented in this study are available on request from the first author.

**Conflicts of Interest:** The authors declare no conflict of interest.

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
