# Peer review of "Does Entrepreneurial Financial Support Guarantee New Ventures’ Performance via Competitive Advantage and Innovation? Empirical Answers from Ho Chi Minh City Region, Vietnam"

_sustainability, doi:10.3390/su152115519_

Round 1

Reviewer 1 Report

I found the article interesting. Significant scientific issues are discussed in the article.

The literature review present theoretical background for the empirical research.

The findings in the discussion section are critically assessed and compared to findings of other authors.

The conclusion provide a neat summary of the main results, practical contribution, research limitations and future research directions.

However, there are some questions and comments that I would like to suggest:

-      what was the procedure for selecting companies for the survey;

-      over what period the research was carried out;

-      it is worth showing the survey questionnaire, which can also be used for comparative analysis for other researchers in the future;

-      the question of whether other authors have done systematic review in this area of research is worth answering.

Author Response

For research article

Response to Reviewer 1 Comments

1. Summary

Thank you very much for taking the time to review this manuscript. We would like to express our gratitude on your valuable comments. We have revised our manuscript according to your recommendations. Please find the detailed responses below and the corresponding revisions/corrections highlighted in the re-submitted files.

2. Questions for General Evaluation

Reviewer’s Evaluation

Response and Revisions

Is the content succinctly described and contextualized with respect to previous and present theoretical background and empirical research (if applicable) on the topic?

Yes

Thanks for your positive feedback

Are all the cited references relevant to the research?

Yes

Thanks for your positive feedback

Are the research design, questions, hypotheses and methods clearly stated?

Yes

Thanks for your positive feedback

Are the arguments and discussion of findings coherent, balanced and compelling?

Yes

Thanks for your positive feedback

For empirical research, are the results clearly presented?

Can be improved

Thanks to your feedback, we have revised our results to depict our differentiation from previous studies and our novelty

Is the article adequately referenced?

Yes

Thanks for your positive feedback

Are the conclusions thoroughly supported by the results presented in the article or referenced in secondary literature?

Can be improved

Thanks for your feedback. We have summarized the important results of our paper which contributed to the literature

3. Point-by-point response to Comments and Suggestions for Authors

Comment 1: I found the article interesting. Significant scientific issues are discussed in the article.

The literature review present theoretical background for the empirical research.

The findings in the discussion section are critically assessed and compared to findings of other authors.

The conclusion provide a neat summary of the main results, practical contribution, research limitations and future research directions.

Response 1: Thanks for your positive comments. We have also made various improvements in our manuscript according to your recommendations.

Comment 2: However, there are some questions and comments that I would like to suggest:

-      what was the procedure for selecting companies for the survey;

Response 2: Thank you for pointing this out. We have provided our procedure of seeking and selecting appropriate new ventures in the “Sampling strategies, sampling design and data collection” section (page 11-12).

Comment 3:

-      over what period the research was carried out;

Response 3: Thanks to your comment, we have clarified the period that our research was conducted in Section 3.2 as following: “from November 2021 to June 2023. This time range illustrated the circumstances of NVs during COVID-19 and post-COVID-19 pandemic phases” (page 12)

Comment 4:

-      it is worth showing the survey questionnaire, which can also be used for comparative analysis for other researchers in the future;

Response 4: Thanks to your valuable feedback, we have provided our full items of our survey questionnaire in “Table 2. Properties of the constructs” that can be utilized in future studies. (page 14-15).

Comment 5:

-      the question of whether other authors have done systematic review in this area of research is worth answering.

Response 5: Thanks for your helpful suggestion. We totally agree with this comment. We have published some systematic literature reviews in entrepreneurship research field, especially entrepreneurial ecosystem which encompassed entrepreneurial financial support as a crucial domain. Hence, we believed that we had deep understanding on the research guidelines for future study in this research field, which has been provided in those systematic literature reviews. Therefore, we have mentioned them and provided the additional reasons and urgency to conduct this research according to their research guidelines associated in their systematic literature reviews.

4. Response to Comments on the Quality of English Language

Point 1:

Quality of English Language

(x) I am not qualified to assess the quality of English in this paper
( ) English very difficult to understand/incomprehensible
( ) Extensive editing of English language required
( ) Moderate editing of English language required
( ) Minor editing of English language required
( ) English language fine. No issues detected

Response 1:  Thanks for your comments on the quality of English language. We have carefully re-read our manuscript and asked our two colleagues who are fluently in English and have published various articles in reputable journals in order to correct our manuscript to enhance the quality of English language of our manuscript.

5. Additional clarifications

Reviewer 2 Report

The topic is very interesting and is well contextualized.

It is also well framed in terms of the supporting theory

The literature review is well done, highlighting current and relevant work.

The research questions and hypotheses are well posed and well founded.

Methodology is appropriate to the type of study.

An interesting discussion is held, highlighting the contribution of the study (theoretical and practical) as well as limitations and possible future work are identified.

It is recommended that fewer acronyms be used. Excessive use makes the study difficult to read.

Author Response

For research article

Response to Reviewer 2 Comments

1. Summary

Thank you very much for taking the time to review this manuscript. We would like to express our gratitude on your valuable comments. We have revised our manuscript according to your recommendations. Please find the detailed responses below and the corresponding revisions/corrections highlighted in the re-submitted files.

2. Questions for General Evaluation

Reviewer’s Evaluation

Response and Revisions

Is the content succinctly described and contextualized with respect to previous and present theoretical background and empirical research (if applicable) on the topic?

Yes

Thanks for your positive feedback

Are all the cited references relevant to the research?

Yes

Thanks for your positive feedback

Are the research design, questions, hypotheses and methods clearly stated?

Yes

Thanks for your positive feedback

Are the arguments and discussion of findings coherent, balanced and compelling?

Yes

Thanks for your positive feedback

For empirical research, are the results clearly presented?

Yes

Thanks for your positive feedback

Is the article adequately referenced?

Yes

Thanks for your positive feedback

Are the conclusions thoroughly supported by the results presented in the article or referenced in secondary literature?

Can be improved

Thanks for your feedback. We have summarized the important results of our paper which contributed to the literature

3. Point-by-point response to Comments and Suggestions for Authors

Comment 1: The topic is very interesting and is well contextualized.

It is also well framed in terms of the supporting theory

The literature review is well done, highlighting current and relevant work.

The research questions and hypotheses are well posed and well founded.

Methodology is appropriate to the type of study.

An interesting discussion is held, highlighting the contribution of the study (theoretical and practical) as well as limitations and possible future work are identified.

It is recommended that fewer acronyms be used. Excessive use makes the study difficult to read.

Response 1: Thanks for your positive comments. We have also made numerous improvements in our manuscript according to your recommendations. In the initial manuscript, we have used various acronyms to keep the length of our paper in an adequate requirement.  However, we acknowledged that the excessive usage of acronyms made our study difficult to read. In the revised version, we have only kept the acronyms of the 5 major factors (Firm financial performance – FiPer, Firm operational performance – BOP, Firm competitive advantage – FCA, firm innovation – INO, and entrepreneurial financial support – FIN) in our conceptual framework and written the other acronyms in proper forms throughout our paper, so that it will be easier to read our research.

4. Response to Comments on the Quality of English Language

Point 1:

Quality of English Language

(x) I am not qualified to assess the quality of English in this paper
( ) English very difficult to understand/incomprehensible
( ) Extensive editing of English language required
( ) Moderate editing of English language required
( ) Minor editing of English language required
( ) English language fine. No issues detected

Response 1:  Thanks for your comments on the quality of English language. We have carefully re-read our manuscript and asked our two colleagues who are fluently in English and have published various articles in reputable journals in order to correct our manuscript to enhance the quality of English language of our manuscript.

5. Additional clarifications

Reviewer 3 Report

1.      The purpose and reason of the research need to be conveyed in more detail

2.      The author has written down some theoretical conditions from relevant and current sources, but still has not presented a picture of the empirical conditions that make gab research and become the idea of this research.

3.      It is worth showing the urgency of research and novelty

4.      The research question must be clear

5.      The theory used has shown and referred to several relevant research results and from reputable journals.

6.      The description of the findings is not enough. All that is needed is to refer back to the results of other studies, highlighting their differences and contributions.

7.      The reasons for the selection of subjects and the location of the study.

8.      The sample selection technique should be clarified again

9.      Reasons for choosing PLS-SEM for this study.

10.  The interpretation of research results is conveyed in more detail, and refers to the strengthening of several supporting theories. And delivered implementation in the field (real conditions).

11.  The conclusion has been well delivered. It is expected that this conclusion item is an answer to the findings of the existing research objectives . Researchers have conveyed the practical implications of the results of this study  in more  detail and more operationally.

12.  It is necessary to convey the uniqueness and differentiation of this study with previous research.

13.  There are no limitations of detailed research.

14.  It is necessary to do professional reading to translate languages

OK

Author Response

For research article

Response to Reviewer 3 Comments

1. Summary

Thank you very much for taking the time to review this manuscript. We would like to express our gratitude on your valuable comments. We have revised our manuscript according to your recommendations. Please find the detailed responses below and the corresponding revisions/corrections highlighted in the re-submitted files.

2. Questions for General Evaluation

Reviewer’s Evaluation

Response and Revisions

Is the content succinctly described and contextualized with respect to previous and present theoretical background and empirical research (if applicable) on the topic?

Can be improved

Thanks to your feedback, we have made several improvements according to your comments

Are all the cited references relevant to the research?

Can be improved

Thanks for your feedback. We have checked our cited references so that they are all related to the research

Are the research design, questions, hypotheses and methods clearly stated?

Can be improved

Thanks to your feedback, we have made several improvements according to your comments

Are the arguments and discussion of findings coherent, balanced and compelling?

Can be improved

Thanks to your feedback, we have improved our “Discussion” section.

For empirical research, are the results clearly presented?

Can be improved

Thanks to your feedback, we have added improved information to make explicit of our results

Is the article adequately referenced?

Can be improved

Thanks for your feedback. We have checked our references so that they are all related to the research

Are the conclusions thoroughly supported by the results presented in the article or referenced in secondary literature?

Can be improved

Thanks for your feedback. We have summarized the important results of our paper which contributed to the literature

3. Point-by-point response to Comments and Suggestions for Authors

Comment 1: The purpose and reason of the research need to be conveyed in more detail.

Response 1: Thank you for pointing this out. We agree with this comment. Therefore, we have provided both practical and theoretical gaps which depict our reasons and purposes in more detail for conducting this research in the “Introduction” section, which will be displayed in the response 2 and 3 (page 1-5).

Comment 2: The author has written down some theoretical conditions from relevant and current sources, but still has not presented a picture of the empirical conditions that make gab research and become the idea of this research.

Response 2: Thanks to your valuable comments, we have provided the practical gaps in the new ventures’ performance, which are the consequences and can be traced back to the problems of entrepreneurial financial support, firm innovation, and firm competitive advantages embedded in Vietnamese entrepreneurship context, demonstrating the practical gaps and purposes of conducting our research (page 1-2).

Comment 3: It is worth showing the urgency of research and novelty.

Response 3: Thanks for your helpful recommendation. Coming along with the urgency of conducting this paper which can be found in the practical gaps, in the theoretical context, we have published some systematic literature reviews in entrepreneurship research field, especially entrepreneurial ecosystem which encompassed entrepreneurial financial support as a crucial domain. Hence, we believed that we had deep understanding on the research guidelines for future study in this research field, which has been provided in those systematic literature reviews. Therefore, we have mentioned them and provided the additional reasons, urgency, and novelty (besides the initial reasons) to conduct this research according to their research guidelines associated in their systematic literature reviews in “Introduction” section (page 3-4). Moreover, we have clarified our novelty in the “Discussion” section by providing our differentiation from the previous studies according to your comments.

Comment 4: The research question must be clear.

Response 4: Thanks for your feedback. We assumed that our research questions can cover and encompass our purposes and results. In addition, we have re-written them in a formal manner instead of using acronyms in our research questions to make them clear.

Comment 5: The theory used has shown and referred to several relevant research results and from reputable journals.

Response 5: Thanks for your positive feedback. We have clarified adopted theory and various relevant research results to support our findings.

Comment 6: The description of the findings is not enough. All that is needed is to refer back to the results of other studies, highlighting their differences and contributions.

Response 6: Thanks for your valuable suggestions. Since all of our hypotheses have been supported, we have referred our results to the relevant research findings of previous studies that have been demonstrated in our “Theoretical background and hypotheses development” section. Besides that, we have also added various ideas which highlighted our differences from the previous studies, while concurrently providing our contributions to the literature (page 17-20).

Comment 7: The reasons for the selection of subjects and the location of the study.

Response 7: Thank you for pointing this out. We totally agree with this comment. Therefore, we have clarified our reasons for the selection of our subjects and the location of the study in our “Methodology” section (page 11-12) including the reasons for selection of Ho Chi Minh City region as our study area (We have also mentioned the reasons to select HCMC region in our “Introduction” section), the reasons for selection of the (1) entrepreneurs of (2) new ventures in the HCMC region. Besides that, we have also provided our procedure of seeking and selecting appropriate new ventures in the “Sampling strategies, sampling design and data collection” section (page 11-12).

Comment 8: The sample selection technique should be clarified again.

Response 8: Thanks to your valuable comments, we have made explicit our sample selection technique in our “Methodology” section.

Comment 9: Reasons for choosing PLS-SEM for this study.

Response 9: Thank you for pointing this out. Therefore, we have clarified our reasons for choosing PLS-SEM in our research according to reputable sources, which can be found at the corrections highlighted in “Analysis” section (page 12).

Comment 10: The interpretation of research results is conveyed in more detail, and refers to the strengthening of several supporting theories. And delivered implementation in the field (real conditions).

Response 10: Thanks for your positive feedback. We have also improved the interpretation of our research results in “Discussion” sections (page 17-23) based on your comments.

Comment 11: The conclusion has been well delivered. It is expected that this conclusion item is an answer to the findings of the existing research objectives. Researchers have conveyed the practical implications of the results of this study in more detail and more operationally.

Response 11: Thanks for your positive feedback.  We have also summarized the important results of our paper which contributed to the literature.

Comment 12: It is necessary to convey the uniqueness and differentiation of this study with previous research.

Response 12: Thanks to your suggestions, we have explained the uniqueness and differentiation of our studies by offering our differences from the previous studies, which can be found in our “Discussion” section (page 17-23), especially “the comprehensive relationships between FIN and new ventures’ firm performance through both the mediating roles of FCA and INO”. Besides that, we have also provides our novelty (in Response 3, the fourth paragraph of our “Introduction” section, and “Conclusion” section)

Comment 13: There are no limitations of detailed research.

Response 13: Thanks to your helpful feedback, we have adjusted our “Limitations and Future Research” section (page 22-23), which displays the limitations of our detailed research, while concurrently offering the research guideline for future studies to enhance of research results.

Comment 14: It is necessary to do professional reading to translate languages.

Response 14: Thanks for your comments on the quality of English language. We have carefully re-read our manuscript and asked our two colleagues who are fluently in English and have published various articles in reputable journals in order to correct our manuscript to enhance the quality of English language of our manuscript.

Comment 15: It is necessary to attach information from the research ethics team.

Response 15: We have also attached information from the research ethics team in our manuscript.

4. Response to Comments on the Quality of English Language

Point 1:

Quality of English Language: OK

( ) I am not qualified to assess the quality of English in this paper
( ) English very difficult to understand/incomprehensible
(x) Extensive editing of English language required
( ) Moderate editing of English language required
( ) Minor editing of English language required
( ) English language fine. No issues detected

Response 1:  Thanks for your comments on the quality of English language. We have carefully re-read our manuscript and asked our two colleagues who are fluently in English and have published various articles in reputable journals in order to correct our manuscript to enhance the quality of English language of our manuscript.

5. Additional clarifications

Reviewer 4 Report

The research applies the resource-based view (RBV) theory to examine the effects of entrepreneurial financial support on new ventures' performance via competitive advantage and innovation. It provides empirical evidence that entrepreneurial financial support has a strong direct and indirect effect on firm innovation and competitive advantage, which in turn mediate the relationships between financial support and firm financial and operational performance. The paper is well-organized and meticulously written. However, I cannot see how this study is related to the aim and scope of the journal. I see the sustainable FCA, for instance, but I do not think that it is not an issue relating to sustainability as defined by the journal itself. I would advise how this research contributes to a sustainable world or economy. Firm innovation should be a means of sustainable development, but I do not see the relationship in the paper at all. After a minor revision, I would be happy to reevaluate the paper.

Author Response

For research article

Response to Reviewer 4 Comments

1. Summary

Thank you very much for taking the time to review this manuscript. We would like to express our gratitude on your valuable comments. We have revised our manuscript according to your recommendations. Please find the detailed responses below and the corresponding revisions/corrections highlighted in the re-submitted files.

2. Questions for General Evaluation

Reviewer’s Evaluation

Response and Revisions

Is the content succinctly described and contextualized with respect to previous and present theoretical background and empirical research (if applicable) on the topic?

Yes

Thanks for your positive feedback

Are all the cited references relevant to the research?

Yes

Thanks for your positive feedback

Are the research design, questions, hypotheses and methods clearly stated?

Yes

Thanks for your positive feedback

Are the arguments and discussion of findings coherent, balanced and compelling?

Yes

Thanks for your positive feedback

For empirical research, are the results clearly presented?

Yes

Thanks for your positive feedback

Is the article adequately referenced?

Yes

Thanks for your positive feedback

Are the conclusions thoroughly supported by the results presented in the article or referenced in secondary literature?

Yes

Thanks for your positive feedback

3. Point-by-point response to Comments and Suggestions for Authors

Comment 1: The research applies the resource-based view (RBV) theory to examine the effects of entrepreneurial financial support on new ventures' performance via competitive advantage and innovation. It provides empirical evidence that entrepreneurial financial support has a strong direct and indirect effect on firm innovation and competitive advantage, which in turn mediate the relationships between financial support and firm financial and operational performance. The paper is well-organized and meticulously written. However, I cannot see how this study is related to the aim and scope of the journal. I see the sustainable FCA, for instance, but I do not think that it is not an issue relating to sustainability as defined by the journal itself. I would advise how this research contributes to a sustainable world or economy. Firm innovation should be a means of sustainable development, but I do not see the relationship in the paper at all. After a minor revision, I would be happy to reevaluate the paper.

Response 1: We would like to express our gratitude on your positive comments. Moreover, thank you for pointing this out. We have made various improvements in our manuscript according to your recommendations. We have written various additional ideas which depicted the links of our results to the issues relating to sustainable world and economic, ensuring that this research is associated with the aim and scope of your journal. We have provided those links throughout our research, especially in the “Introduction” and “Practical implication” sections that demonstrated the reasons, purposes, and results of our research related to sustainable development or sustainability.

4. Response to Comments on the Quality of English Language

Point 1:

Quality of English Language

(x) I am not qualified to assess the quality of English in this paper
( ) English very difficult to understand/incomprehensible
( ) Extensive editing of English language required
( ) Moderate editing of English language required
( ) Minor editing of English language required
( ) English language fine. No issues detected

Response 1:  Thanks for your response to comments on the quality of English language. We have carefully re-read our manuscript and asked our two colleagues who are fluently in English and have published various articles in reputable journals in order to correct our manuscript to enhance the quality of English language of our manuscript.

5. Additional clarifications
